# Anthracycline-Induced Subclinical Right Ventricular Dysfunction in Breast Cancer Patients: A Systematic Review and Meta-Analysis

**DOI:** 10.3390/cancers16223883

**Published:** 2024-11-20

**Authors:** Andrea Faggiano, Elisa Gherbesi, Chiara Giordano, Giacomo Gamberini, Marco Vicenzi, Cesare Cuspidi, Stefano Carugo, Carlo M. Cipolla, Daniela M. Cardinale

**Affiliations:** 1Department of Cardio-Thoracic-Vascular Diseases, Foundation IRCCS Ca’ Granda Ospedale Maggiore Policlinico, 20122 Milan, Italymarco.vicenzi@unimi.it (M.V.);; 2Department of Clinical Sciences and Community Health, University of Milan, 20122 Milan, Italy; 3Department of Medicine and Surgery, University of Milano-Bicocca, 20125 Milan, Italy; 4Cardioncology Unit, Cardioncology and Second Opinion Division, European Institute of Oncology, I.R.C.C.S., 20141 Milan, Italydaniela.cardinale@ieo.it (D.M.C.)

**Keywords:** anthracycline chemotherapy, right ventricular dysfunction, breast cancer, strain echocardiography, cardiotoxicity

## Abstract

Anthracycline chemotherapy is essential in treating breast cancer, yet it can lead to heart damage, particularly affecting the left ventricle. Less is known about its impact on the right ventricle. This meta-analysis examines how anthracyclines may cause subclinical damage to the right ventricle in breast cancer patients, as shown by both advanced strain parameters and traditional echocardiographic measures. Analyzing data from 15 studies and nearly 1200 patients, we found significant declines in right ventricular function post-treatment. Interestingly, this subclinical dysfunction does not appear linked to left ventricle damage or to higher chemotherapy doses, suggesting a unique mechanism for right ventricular impairment. These insights underscore the potential benefit of monitoring RV function in patients undergoing anthracycline treatment, as early detection may lead to improved patient care.

## 1. Introduction

Anthracycline chemotherapy remains a cornerstone in the treatment of breast cancer due to its efficacy [1]. However, the well-documented cardiotoxic effects of anthracyclines, particularly in causing left ventricular (LV) dysfunction, pose a significant clinical challenge [2,3]. These drugs, including doxorubicin (DOX) and Epirubicin (EPI), have been extensively studied for their deleterious impact on left ventricular ejection fraction (LVEF) and LV global longitudinal strain (LV GLS), with established correlations to subsequent heart failure and negative prognosis [4,5]. Despite the robust body of literature on LV dysfunction, emerging evidence suggests that the right ventricle (RV) may also be affected by anthracycline-induced cardiotoxicity, yet this has received comparatively less attention, especially in breast cancer patients [6,7,8].

Subclinical RV systolic dysfunction, detected by reduced RV global longitudinal strain (RV GLS) and RV free-wall longitudinal strain (RV FWLS), may precede overt clinical symptoms, highlighting the importance of early detection. Advanced echocardiographic techniques, such as two-dimensional (2D) and three-dimensional (3D) speckle tracking echocardiography (STE), allow for sensitive and detailed assessment of myocardial mechanics, offering an opportunity to evaluate RV function more precisely [9,10,11,12]. While previous studies have primarily focused on the LV [13,14], the potential for anthracyclines to impair RV systolic function remains underexplored. Likewise, it is unclear whether the damage to the RV is a direct result of anthracycline-induced myocardial injury or, possibly, is mediated indirectly through LV dysfunction due to ventricular interdependence. Understanding the extent and nature of this dysfunction is critical for comprehensive cardiac monitoring and could influence treatment decisions and long-term patient outcomes. Therefore, this systematic review and meta-analysis aim to explore the effect of anthracycline chemotherapy on RV systolic function in breast cancer patients, as evaluated through echocardiographic strain analysis (RV GLS and RV FWLS).

## 2. Methods

### 2.1. Search and Study Selection

This research was conducted in accordance with the Preferred Reporting Items for Systematic Reviews and Meta-Analyses (PRISMA) guidelines [15], and the systematic review was registered on the International Prospective Register of Systematic Reviews (PROSPERO) with the identifier CRD42024591588 [16]. The relevant literature was systematically reviewed to identify all studies investigating the effect of anthracycline chemotherapy in breast cancer patients on subclinical RV dysfunction, assessed using RV GLS and RV FWLS via 2D and 3D speckle tracking echocardiography (STE). The PubMed, OVID-MEDLINE, and Cochrane Library databases were searched for English-language articles published from inception to 9 September 2024. Studies were identified using MeSH terms and by combining the following keywords: “breast cancer”, “anthracycline”, “doxorubicin”, “chemotherapy”, “right ventricular systolic dysfunction”, “right ventricle global longitudinal strain”, “right ventricle free-wall longitudinal strain”, “right ventricular mechanics”, “speckle-tracking echocardiography”, and “strain analysis echocardiography”. The electronic search was supplemented by manual checks of reference lists from selected papers. Reviews, editorials, and case reports were excluded from the analysis but were screened for potential additional references. Two authors (C.G. and G.G.) independently assessed the abstracts and full texts of the retrieved studies to determine eligibility based on the inclusion criteria outlined below. A third reviewer (A.F.) resolved any disagreements regarding study eligibility. Data extraction was performed by one reviewer (A.F.) and independently verified by another (E.G.).

The main inclusion criteria were: (I) Articles published in peer-reviewed journals; (II) studies reporting echocardiographic data on RV mechanics (RV GLS and/or RV FWLS) before and after anthracycline chemotherapy in breast cancer patients; (III) availability of a minimum set of clinical and demographic data; (IV) a follow-up duration of more than one month.

Specific exclusion criteria included: (I) studies with fewer than ten patients; (II) studies involving children or adolescents (age < 18 years); (III) studies in which more than 40% of patients had cancers other than breast cancer.

For each eligible study, data such as article details, study characteristics, relevant population demographics, and echocardiographic data were systematically extracted. Where essential data were unavailable, the corresponding author was contacted, and if data could not be obtained, they were extrapolated from the figures in the studies. When multiple follow-up assessments were available in the studies, the echocardiographic data from the final evaluation, representing the longest follow-up period, were used for the meta-analysis. When quantitative variables were presented as medians and confidence intervals (CIs), the means and standard deviations required for meta-analysis were calculated using appropriate formulas [17,18]. In studies where the specific type of anthracycline was not indicated, it was reasonably inferred from the cumulative dosage. This inference was indicated in the tables with the term “probably” to reflect the uncertainty associated with the identification of the specific anthracycline. The cumulative DOX-equivalent dose for studies using EPI was determined according to specific conversion factors (https://www.cancercalc.com/anthracycline.php (accessed on 31 October 2024)) [19,20].

The Newcastle-Ottawa Scale (NOS) was used to assess the quality of nonrandomized studies in meta-analyses (http://www.ohri.ca/programs/clinical_epidemiology/oxford.htm (accessed on 5 October 2024)) [21]. All analyses were based on previously published studies; thus, no ethical approval or patient consent was required.

### 2.2. Statistical Analysis

The primary outcome of the meta-analysis was to evaluate changes in RV GLS and RV FWLS induced by anthracycline chemotherapy in breast cancer patients. Secondary outcomes included changes in other traditional echocardiographic parameters of RV function: Tricuspid Annular Plane Systolic Excursion (TAPSE), Fractional Area Change (FAC), and Tissue Doppler Imaging Systolic Velocity (TDI S’). To this end, a pooled analysis of cardiac parameters was conducted using fixed or random effects meta-analysis with Comprehensive Meta-Analysis Version 2, Biostat, Englewood, NJ, USA. Standardized mean differences (SMD) with 95% CIs were used to evaluate the statistical differences in RV function parameters before and after anthracycline chemotherapy. Statistical significance was set at *p* < 0.05. Heterogeneity was assessed using I-square, Q, and tau-square statistics; random or fixed effect models were applied based on study heterogeneity (I^2^) [22,23]. Publication bias was evaluated using funnel plots according to the trim-and-fill method and observed and adjusted values with their respective lower and upper limits were calculated [24,25]. To assess the impact of individual studies on the overall results, a sensitivity analysis was performed by excluding each study one by one and recalculating the combined estimates from the remaining studies [26,27]. Meta-regressions were conducted to synthesize research findings while adjusting for the effects of available covariates on the response variable [28].

## 3. Results

### 3.1. Characteristics of the Included Studies

After removing duplicates, the initial literature search identified 543 papers. The PRISMA flowchart showing the search strategy and manuscript selection process is illustrated in Figure 1.

After the initial screening of titles and abstracts, 431 studies were excluded as they were not related to the topic. Consequently, 112 studies were fully reviewed; of these, 35 did not report data on RV myocardial mechanics (RV GLS or RV FWLS) before and after chemotherapy, 18 referred to chemotherapy regimens other than anthracycline, 23 referred to tumors other than breast cancer (with >40% of patients having different tumors), 20 were reviews, commentaries, or editorials. According to the NOS, the quality of the studies ranged from 6 to 9 (i.e., a score that identifies studies of fair or good quality). Therefore, no study was excluded based on its limited quality (Appendix A).

A total of 1148 patients who underwent anthracycline treatment for breast cancer were included in 15 studies, with sample sizes ranging from 28 to 351 [29,30,31,32,33,34,35,36,37,38,39,40,41,42,43]. These studies were conducted across five continental regions: Asia (eight studies), Europe (three studies), Africa (two studies), North America (one study), and South America (one study).

Table 1 summarizes the key findings from the selected studies, including the authors, year of publication, breast cancer type, anthracycline used, concomitant anti-Human Epidermal Growth Factor Receptor 2 (HER2) therapy and/or radiotherapy, sample size, gender distribution, and the main LV echocardiographic parameters.

With the exception of two studies (Attar et al. [36], Ghaznawie et al. [39]), which included patients with cancers other than breast cancer (though in both studies, over 65% of the patients had breast cancer), all remaining studies focused exclusively on breast cancer patients, including both HER2-positive and HER2-negative cases. Only a few studies provided details regarding the disease stage, specifically indicating whether patients with metastatic disease were excluded. DOX and EPI were used equally as the primary anthracyclines in treatment. Details of the chemotherapy regimen are provided in Appendix A, showing a pooled cumulative DOX-equivalent dose of 231.8 ± 4.7 mg/m^2^. In four out of 15 studies, a subset of patients (ranging from 6% to 76%) also received concomitant radiotherapy, while four studies included patients receiving concurrent anti-HER2 therapy (e.g., trastuzumab, with use ranging from 18% to 82%). As expected, the majority of patients were female, with a pooled mean age of 48 ± 2 years. Echocardiographic follow-up assessments were predominantly conducted at the completion of the chemotherapy regimen and never before 3 months from the start of treatment. LV and RV myocardial deformation indexes were measured offline from 2D or 3D echocardiographic images using commercial dedicated software; R-R gating was used for strain assessment. In all studies, LV and RV endocardium were manually traced and corrected, if necessary, and the average longitudinal strain curve was automatically provided by the software. Echocardiographic machines from three different manufacturers were utilized across the studies: General Electrics (*n* = 10), Philips (*n* = 2), and Toshiba (*n* = 2), with EchoPAC being the most frequently used off-line software for cardiac mechanics analysis. Five studies employed 3D speckle-tracking echocardiography.

### 3.2. Echocardiographic Findings

As expected and consistent with the literature [44,45,46], the meta-analysis revealed a significant worsening in LVEF and LV GLS values following anthracycline chemotherapy. Specifically, the pooled LVEF values were 63.5 ± 0.8% at baseline and 60.1 ± 1.1% at follow-up (SMD: −0.20 ± 0.03, 95% CI: −0.260 to −0.137, *p* < 0.001), while the pooled LV GLS values were 19.88 ± 0.45% at baseline and 17.41 ± 0.33% at follow-up (SMD: −0.28 ± 0.03, 95% CI: −0.341 to −0.215, *p* < 0.001).

Table 2 provides an overview of the RV echocardiographic changes before and after anthracycline chemotherapy in breast cancer patients, highlighting both myocardial deformation parameters (RV GLS and RV FWLS) and traditional measures of RV function.

Overall, all RV echocardiographic parameters identified a subclinical worsening of RV systolic function after treatment.

#### 3.2.1. RV Mechanics and 3D Analysis

Pre- and post-treatment mean RV GLS values (including both 2D and 3D studies) in the pooled study population (data from 11 studies) ranged from 22.95% to 25.02% at baseline and from 18.88% to 21.82% at follow-up, with average pooled values of 23.99 ± 0.53% and 20.35 ± 0.75%, respectively. Figure 2 illustrates the meta-analysis findings, where SMD indicated a significant worsening in RV GLS following anthracycline treatment (SMD: −0.259 ± 0.032, 95% CI: −0.322 to −0.196, *p* < 0.0001). This decline was slightly more pronounced in the 3D RV GLS evaluation, with an SMD of −0.337 ± 0.056 (95% CI: −0.446 to −0.299, *p* < 0.001, data from five studies). Similarly, the meta-analysis documented a significant change of similar magnitude in RV FWLS after anthracycline chemotherapy (SMD: −0.269 ± 0.033, 95% CI: −0.335 to −0.203, *p* < 0.001, Figure 3), with pooled RV FWLS values (data from 11 studies) of 24.92 ± 0.48% at baseline and 21.56 ± 0.92% at the end of follow-up. Finally, 3D echocardiographic volume analysis revealed a slight but statistically significant reduction in RVEF, from 55.84 ± 1.88% to 51.88 ± 1.09% post-chemotherapy (SMD: −0.195 ± 0.065, 95% CI: −0.322 to −0.068, *p* = 0.003, data from three studies).

#### 3.2.2. Traditional Parameters of Right Ventricular Function

Among the traditional echocardiographic parameters of RV function, FAC showed the best performance in detecting subclinical deterioration. Specifically, pooled FAC values (data from nine studies) were 50.50 ± 1.66% at baseline and 46.90 ± 1.52% post-chemotherapy, with an SMD of −0.265 ± 0.044 (95% CI: −0.352 to −0.178, *p* < 0.001, Figure 4a). Similarly, the meta-analysis documented a significant, though slightly less pronounced, decline in longitudinal function as assessed by TAPSE after anthracycline chemotherapy (SMD: −0.203 ± 0.045, 95% CI: −0.292 to −0.115, *p* < 0.001, Figure 4b), with pooled TAPSE values (data from nine studies) of 19.24 ± 2.72 mm at baseline and 17.52 ± 3.22 mm at the end of follow-up.

Finally, TDI S’ showed the smallest, albeit still significant, difference pre- and post-chemotherapy. Pooled mean TDI S’ values decreased slightly from 15.05 ± 1.26 cm/s to 14.63 ± 1.46 cm/s, with an SMD of −0.144 ± 0.041 (95% CI: −0.224 to −0.064, *p* < 0.001, data from 10 studies, Figure 4c). Although still within normal limits, the observed subclinical decline in systolic function was accompanied by a mild, clinically insignificant increase in pulmonary artery systolic pressure (PAPS), from 19.96 ± 3.58 mmHg to 20.15 ± 4.13 mmHg (SMD: 0.193 ± 0.072, 95% CI: 0.052 to 0.335, *p* = 0.007, data from three studies).

### 3.3. Publication Bias

Sensitivity analysis indicated a single-study effect only in the case of 3D RVEF; specifically, when the Rossetto study [43] was excluded, the difference between pre- and post-chemotherapy values was no longer statistically significant (SMD: −0.108 ± 0.079, 95% CI: −0.263 to 0.047, *p* = 0.173). For all other echocardiographic parameters analyzed, the presence of a single-study effect was excluded, and no significant publication bias was detected in studies reporting RV function parameters before and after anthracycline chemotherapy. The differences between pre- and post-chemotherapy values remained significant even after correcting for publication bias:RV GLS: SMD: −0.170 (95% CI: −0.224 to −0.116) (Appendix A).RV FWLS: SMD: −0.182 (95% CI: −0.24 to −0.124) (Appendix A).TAPSE: SMD: −0.185 (95% CI: −0.271 to −0.099).FAC: SMD: −0.311 (95% CI: −0.391 to −0.232).PAPS: SMD: 0.140 (95% CI: 0.025 to 0.255).

### 3.4. Correlation Analyses

Given the development of concomitant subclinical LV dysfunction, it was hypothesized that the observed RV subclinical dysfunction might not result from direct cardiotoxic damage caused by anthracyclines but could instead be secondary to the effects of LV dysfunction. To explore this, a meta-regression analysis was conducted between post-chemotherapy LVEF and the effect sizes of RV GLS and RV FWLS (expressed as SMD), aiming to assess the influence of traditional LV function on the occurrence of subclinical RV dysfunction. The analysis did not reveal a significant relationship between the SMD of RV GLS or RV FWLS and LVEF (coefficient: 0.012, *p* = 0.329 and coefficient: 0.015, *p* = 0.137, respectively). Similarly, as shown in Figure 5, lower post-chemotherapy LV-GLS values did not correlate with the occurrence of subclinical RV dysfunction (coefficient for RV GLS: −0.005, *p* = 0.882, Figure 5a; and coefficient for RV FWLS: −0.002, *p* = 0.950, Figure 5b). Finally, we hypothesized that subclinical RV damage, like LV cardiotoxicity, might be dose-dependent. To investigate this, a meta-regression analysis was performed between the cumulative DOX-equivalent dose and the effect sizes of RV GLS and RV FWLS (expressed as SMD), with the aim of evaluating the influence of anthracycline dose on the incidence of subclinical RV dysfunction. The analysis did not reveal a significant relationship between the SMD of RV GLS and RV FWLS and the cumulative DOX-equivalent dose (coefficient: −0.0002, *p* = 0.488 for RV GLS; coefficient: 0.0242, *p* = 0.876 for RV FWLS, respectively).

## 4. Discussion

This meta-analysis provides important insights into the effects of anthracycline chemotherapy on RV systolic function in breast cancer patients. Across the 16 studies reviewed, we observed a consistent decline in RV function following anthracycline treatment. This was evidenced by significant changes in both advanced strain measurements—such as RV GLS and RV FWLS—and traditional echocardiographic parameters, including TAPSE, FAC, and S’. While all the methods identified a deterioration in RV function, strain analysis—particularly the 3D strain— may be more sensitive in detecting the myocardial damage caused by anthracyclines, capturing more pronounced alterations in RV function, even though all values remained within accepted normal limits. In comparison with the meta-analyses by Shi et al. [6] and Kariyanna et al. [7], which focused on broader cancer populations, our study is specifically centered on breast cancer patients, a group with heightened susceptibility to cardiotoxicity due to the cumulative effects of anthracyclines, often combined with radiotherapy or HER2-targeted therapies [47,48].

Notably, one of the key challenges in cardio-oncology is the early detection of subclinical cardiotoxicity [49]. If we could reliably identify cardiac damage at an earlier stage, we could implement timely therapeutic strategies, such as cardioprotective agents, to prevent progression to overt heart failure [2,50,51]. Novel approaches, including the use of SGLT2 inhibitors, have shown promise in offering cardiac protection and could be explored further in this context [52,53,54]. The question, therefore, is whether RV strain analysis adds value in detecting subclinical cardiotoxicity earlier and more reliably than traditional methods.

In this meta-analysis, traditional echocardiographic parameters, such as TAPSE, FAC, and S’, also detected small but significant declines in RV function, which raises the question of why RV strain should be favored. The answer may lie in the superior precision and sensitivity of strain analysis. Indeed, although the pooled values of traditional RV parameters in our meta-analysis indicate a post-chemotherapy decline, this reduction is not consistently observed in smaller, individual studies. Specifically, in six out of nine studies, TAPSE did not show significant reductions and similar inconsistencies were noted for FAC (non-significant in four out of nine studies) and TDI-S’ (non-significant in eight out of ten studies). In contrast, all but two studies (El-Sherbeny et al. [38] for RV FWLS and Giang et al. [40] for RV GLS) reported significant declines in parameters related to RV myocardial mechanics. These findings highlight the potential of RV strain analysis to detect subtle myocardial changes that traditional echocardiographic measures might miss. However, in a setting where both traditional and strain-based methods reveal a subclinical deterioration, the added value of routinely using RV strain becomes less clear. This is particularly important given the additional time, specialized software, and optimal acoustic windows required for strain imaging, which may not be warranted unless its unique prognostic value or ability to guide early intervention is clearly demonstrated [55,56].

Unlike in other clinical settings [57,58,59], the question of whether RV strain has a distinct prognostic value remains unanswered here. Although strain analysis provides a detailed view of myocardial mechanics, no studies in our meta-analysis conclusively demonstrated that subclinical RV dysfunction, as detected by RV strain parameters, precedes or predicts overt LV dysfunction. In fact, studies like Rossetto et al. [43] and Boczar et al. [29] suggested that RV and LV dysfunctions occur concurrently, with no evidence of temporal precedence. As noted by Attar et al. [36], while RV strain can detect early dysfunction, its ability to predict progression to clinical cardiotoxicity or serve as a marker for future adverse outcomes remains unclear. Interestingly, Chang et al. [30] reported that RV FWLS represents an independent predictor of dyspnoea in breast cancer patients treated with anthracycline, independently of LV and RV systolic and diastolic function.

The observed subclinical RV dysfunction raises a critical point: whether it represents a direct consequence of anthracycline-induced insult or if it is a hemodynamic consequence of LV dysfunction [60]. Even if none of the included studies specifically reported identifying patients with isolated clinical RV dysfunction, our meta-regression analysis did not find a significant relationship between post-chemotherapy LVEF and changes in RV GLS and RV FWLS. This suggests that RV dysfunction may not be a consequence of impaired LV function due to ventricular interdependence. Interestingly, the difference is not significantly more pronounced for RV FWLS (SMD for RV GLS: −0.259 ± 0.032 vs. −0.269 ± 0.033 for RV FWLS), suggesting that both the RV myocardium and the interventricular septum exhibit similar susceptibility to the direct effects of chemotherapy. Moreover, the reviewed studies do not support the hypothesis that RV dysfunction precedes or predicts LV cardiotoxicity, indicating that both ventricles may experience concurrent but independent damage. Several included studies (e.g., Boczar et al. [29]; Xu et al. [35]) also point toward anthracyclines exerting direct cardiotoxic effects on the RV myocardium, as evidenced by changes in strain parameters. This supports the notion that RV dysfunction can occur independently of LV impairment. Indeed, mechanistically, anthracyclines generate reactive oxygen species that interact with intracellular iron, producing hydroxyl radicals [61,62]. These radicals lead to oxidative stress, which damages cardiomyocytes, promoting fibrosis and cell death, particularly in the RV due to its thinner wall [63,64]. Furthermore, the inhibition of topoisomerase IIβ in cardiomyocytes by anthracyclines is another proposed mechanism for RV failure, leading to DNA damage and apoptosis [65,66]. This direct damage mechanism explains the concurrent RV dysfunction observed alongside LV dysfunction rather than one preceding the other. Interestingly, our meta-regression suggests that RV cardiotoxicity does not appear to be dose-dependent on anthracycline exposure, in contrast to LV cardiotoxicity [67,68]. This supports the hypothesis of a distinct underlying damage mechanism. However, it is important to note that all studies included in our analysis reported cumulative DOX-equivalent doses ranging from 193.1 to 258.4 mg/m^2^—substantially below the critical thresholds (>400–450 mg/m^2^ [69,70,71])—which may limit our capacity to detect significant dose-dependent effects at higher doses. An additional, less-explored aspect is the potential early role of left atrial strain (LAS) in detecting cardiac dysfunction in this setting [72,73,74]. Preliminary data from our meta-analysis on the effects of anthracycline chemotherapy on LAS demonstrated a significant worsening in peak atrial longitudinal strain (30.1% vs. 26.8%, *p* < 0.001) within a year of starting chemotherapy in 343 patients from 10 studies [75]. Similarly, this suggests that anthracyclines can directly impact left atrial function, potentially leading to a condition we might refer to as left atrial cardiomyopathy [76,77]. Alternatively, LAS could serve as a marker of the complex interdependence between the LV and RV, where it reflects both left atrial dysfunction and the compensatory changes occurring in response to the interplay between ventricular preload and afterload. However, recognizing the possibility of an independent mechanism to directly impact the RV ventricle strengthens the rationale for incorporating RV strain analysis into monitoring protocols. Nevertheless, further research is necessary to determine its clinical utility, particularly in predicting long-term outcomes and informing early intervention strategies [78,79]. Given the absence of strong prognostic data, it is important to exercise caution when advocating for the routine implementation of RV strain analysis in clinical practice. While RV strain can detect subtle dysfunction, it remains unclear whether this information leads to improved patient management or outcomes. For example, studies such as those by Attar et al. [36] and Ghaznawie et al. [39] did not provide evidence that subclinical RV dysfunction is associated with an increased risk of adverse outcomes, such as heart failure or mortality. As such, the clinical significance of RV dysfunction remains speculative at this stage [80].

Despite these uncertainties, RV strain analysis may still have a role in more comprehensive cardiac evaluations, particularly given the sensitivity of 3D strain imaging, as shown in our meta-analysis. If future research demonstrates that early detection of subclinical RV dysfunction may lead to changes in treatment strategies—such as the initiation of cardioprotective agents—then RV strain analysis could become a valuable tool in the cardio-oncology setting. Until more robust data is available, routine use of RV strain alongside LV strain should be approached with caution, particularly in resource-limited settings.

In conclusion, while this meta-analysis demonstrates that both traditional echocardiographic methods and strain analysis detect subclinical RV dysfunction after anthracycline chemotherapy in breast cancer patients, the additional clinical value of RV strain remains uncertain. The deterioration in RV function, while technically detectable, generally remains subclinical, with no clear evidence that it predicts adverse outcomes or LV dysfunction. Future research should aim to clarify the prognostic significance of subclinical RV dysfunction and determine whether early detection via RV strain could lead to timely interventions that improve patient outcomes. For now, the routine implementation of RV strain analysis should be considered on a case-by-case basis, with a focus on further validating its role in clinical practice.

## 5. Limitations

Our meta-analysis has several limitations that we would like to address. One of the primary constraints of our study pertains to the inclusion of studies utilizing both 2D and 3D echocardiographic analyses. To mitigate this limitation, as outlined in Section 3, we conducted sensitivity analyses (excluding the 2D studies) to specifically assess the value of 3D speckle-tracking echocardiography.

Furthermore, the included studies employed different echocardiographic equipment and off-line strain analysis software, and these technical differences (outlined in Table 1) may have influenced our findings. A further limitation is related to the inability to provide a pooled follow-up parameter, as some articles did not report the follow-up duration but rather specified the number of chemotherapy cycles administered without indicating the length of each cycle or the interval between cycles. To address this limitation, we included the individual follow-up for each study in Table 1, specifying in the Section 3 that follow-ups were predominantly conducted at the completion of the chemotherapy regimen and never before 3- months from the start of treatment.

Additionally, only a few studies provided comparative results on RV function between those who did and those who did not develop LV cardiotoxicity (dysfunction). As a result, we were unable to perform a targeted meta-analysis on these two subpopulations, which would have helped clarify whether the observed subclinical RV dysfunction is due to direct anthracycline toxicity or secondary to LV dysfunction. However, to address this limitation, we performed meta-regressions, which revealed that post-chemotherapy LVEF and LV GLS did not correlate with the occurrence of subclinical RV dysfunction, suggesting the possibility that anthracyclines may indeed exert direct cardiotoxic effects on the right heart. Unfortunately, only a few studies included subjects who underwent radiotherapy and/or concurrent anti-HER2 therapy and especially did not provide separate data for patients with and without these concomitant therapies. This limitation hindered our ability to assess any potential differential effects of these treatments on the development of subclinical RV dysfunction [81,82].

Finally, nearly none of the studies evaluated the long-term prognostic impact of developing subclinical RV dysfunction on various outcomes, such as subsequent LV dysfunction, and hard outcomes like mortality or heart failure-related hospitalizations. This limitation, along with the fact that a patient-level meta-analysis was not feasible, made it impossible to identify specific RV GLS or RV FWLS thresholds/cut-offs capable of accurately predicting subsequent outcomes.

## 6. Conclusions

In conclusion, this meta-analysis confirms that anthracycline chemotherapy may induce subclinical RV dysfunction in breast cancer patients, as detectable by both traditional echocardiographic parameters and strain-based measures. Although RV strain, particularly 3D strain, may be more sensitive in detecting early myocardial changes, the clinical significance of these subclinical declines remains uncertain. Further research is required to establish the prognostic value of RV strain analysis and its role in guiding early interventions, particularly in preventing progression to overt cardiotoxicity. Until more robust data are available, RV strain analysis should be considered on a case-by-case basis, focusing on its potential to improve early detection and guide interventions while recognizing the need for further validation of its clinical utility.

## Figures and Tables

**Figure 1 cancers-16-03883-f001:**
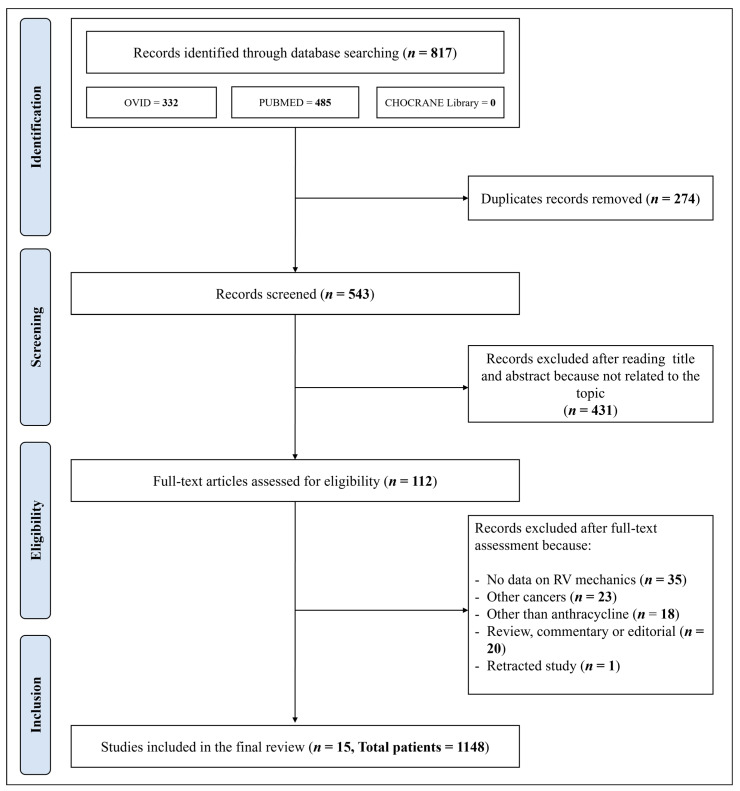
PRISMA flowchart showing the search strategy and manuscript selection process.

**Figure 2 cancers-16-03883-f002:**
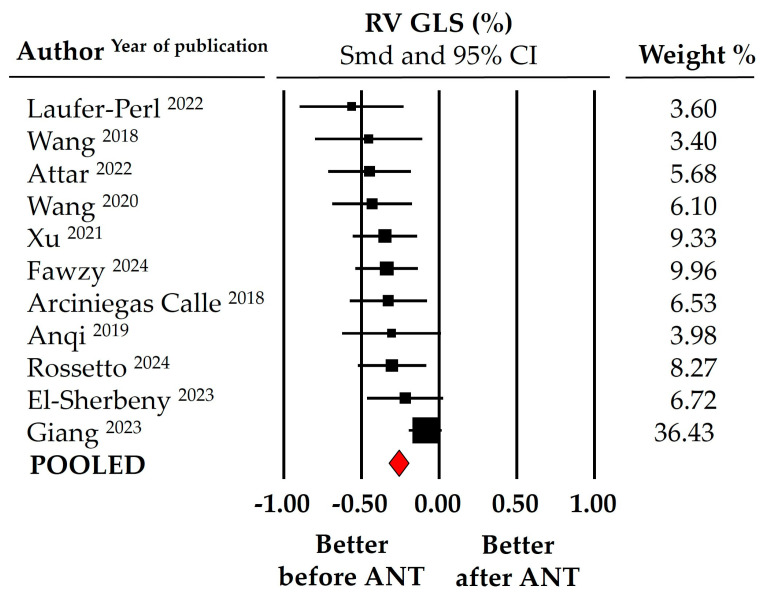
Forest plot for standard means difference (SMD) of right ventricular global longitudinal strain (RV GLS) in patients with breast cancer before and after anthracycline (ANT) chemotherapy [31,32,33,34,35,36,37,38,40,41,43]. The relative weight of each study is reported on the right side. CI: confidence interval.

**Figure 3 cancers-16-03883-f003:**
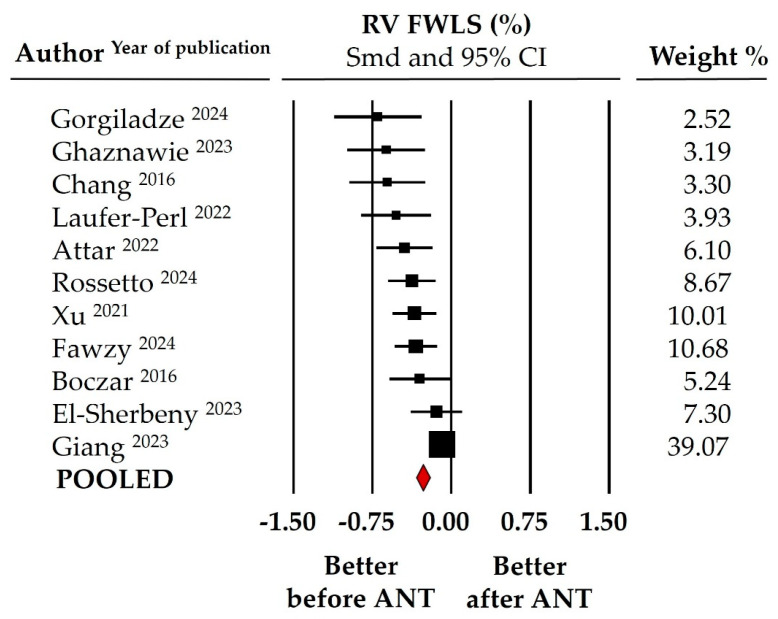
Forest plot for standard means difference (SMD) of right ventricular free-wall longitudinal strain (RV FWLS) in patients with breast cancer before and after anthracycline (ANT) chemotherapy [29,30,35,36,37,38,39,40,41,42,43]. The relative weight of each study is reported on the right side. CI: confidence interval.

**Figure 4 cancers-16-03883-f004:**
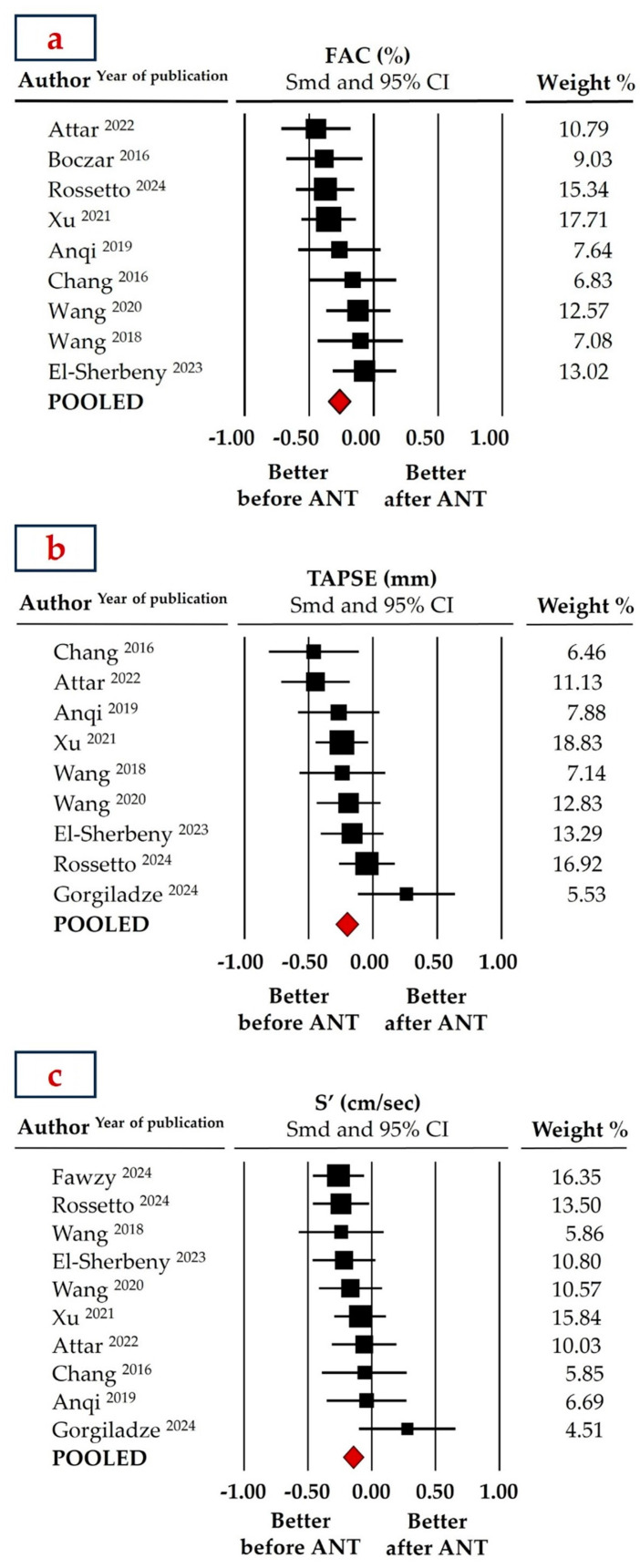
Forest plots for standard means difference (SMD) of Fractional Area Change (FAC, (**a**)) [29,30,32,33,34,35,36,38,43], Tricuspid Annular Plane Systolic Excursion (TAPSE, (**b**)) [30,32,33,34,35,36,37,42,43], and Tissue Doppler Imaging Systolic Velocity (S’, (**c**)) [30,32,33,34,35,36,38,41,42,43] in patients with breast cancer before and after anthracycline (ANT) chemotherapy. The relative weight of each study is reported on the right side. CI: confidence interval.

**Figure 5 cancers-16-03883-f005:**
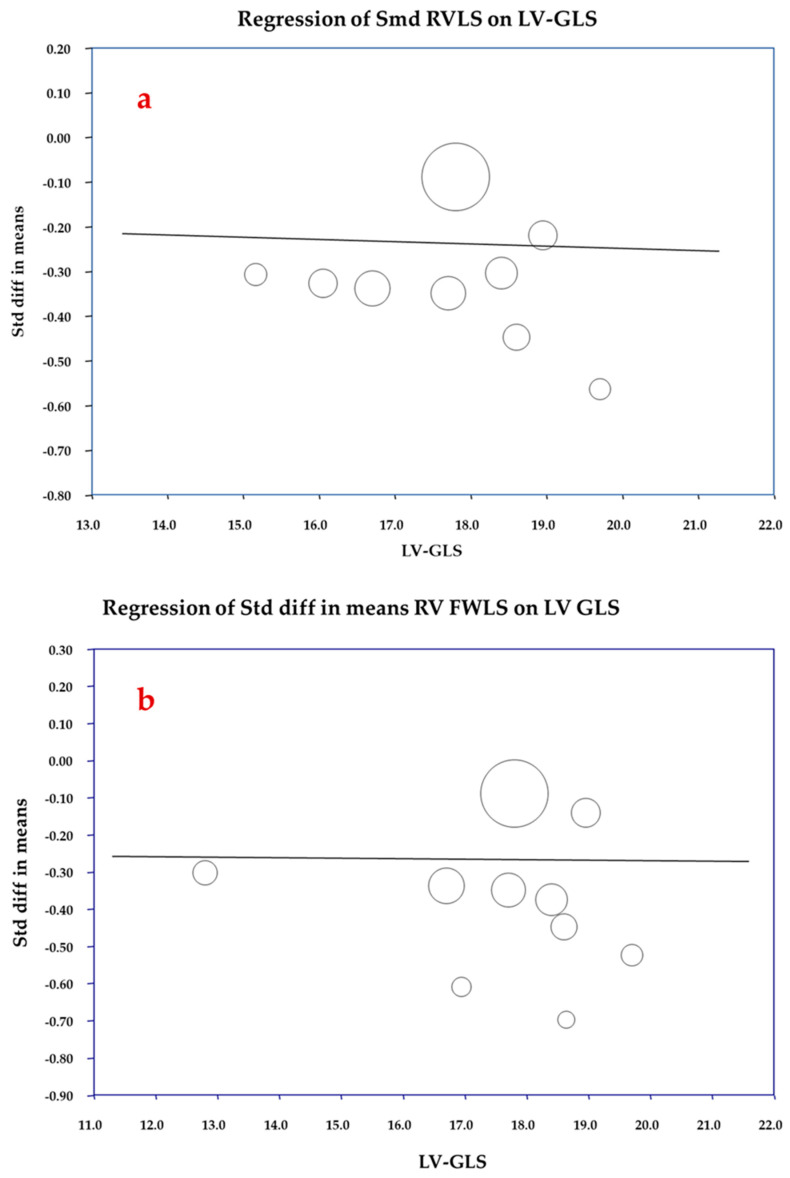
Meta-regressions analysis between left ventricular global longitudinal strain (LV-GLS) and the effect size (standard means difference, SMD) of right ventricular global longitudinal strain (RVGLS, (**a**)) and right ventricular free-wall longitudinal strain (RV FWLS, (**b**)).

**Table 1 cancers-16-03883-t001:** Summary of Characteristics and Main Clinical Variables of the Studies Included in the Systematic Review and Meta-Analysis. ANT: Anthracycline, HER2: human epidermal growth factor receptor 2, SD: standard deviation, LVEF: left ventricle ejection fraction, LV-GLS: left ventricle global longitudinal strain, CT: chemotherapy, 2D: two-dimension, 3D: three-dimension, EPI: Epirubicin, DOX: Doxorubicin, PIRA: Pirarubicin, N.A.: not-available, GE: General Electric.

Author, Publication Year	Breast Cancer Type	ANT Drug Name	Concomitant Anti-HER2	Concomitant Radiotherapy	Sample Size	Age (SD)	Female (%)	Follow-Up	Echocardiography Type	LVEF (%)		LV-GLS (%)	
										Before	After	Before	After
Boczar, 2016 [29]	Early-stage, HER2 negative	DOX or EPI	No	No	49	53 ± 3	98	End of CT regimen (≅ 4 months)	2D TomTec Software	62 ± 7	57 ± 8	15.4 ± 3.6	12.8 ± 4.9
Chang, 2016 [30]	N.A.	Probably EPI	No	6%	35	45 ± 9	100	End of 3-cycle CT regimen (≅ 2 months	2D, GE Vivid E9 EchoPAC Software	68 ± 4	66 ± 9	21.4 ± 4.1	16.9 ± 6.8
Arciniegas Calle, 2018 [31]	Early-stage, HER2 positive	DOX or EPI	82%	76%	66	52 ± 9	100	End of 2-cycle CT regimen (≅ 5 months)	2D, GE Vivid 7	65 ± 4	61 ± 4	17.8 ± 3.6	16.1 ± 3.4
Wang, 2018 [32]	Invasive	PIRA	No	No	36	34 ± 7	100	End of 6-cycle CT regimen (≅ 6 months)	3D, Toshiba Artida SSH-880CV	67 ± 3	67 ± 2	N.A.	N.A.
Anqi, 2019 [33]	N.A.	Probably EPI	No	No	40	47 ± 10	100	End of 6-cycle CT regimen (≅ 6 months)	2D, Philips EPIQ7C QLAB10.8 Software	70 ± 6	60 ± 5	19.7 ± 2.5	15.2 ± 2.1
Wang, 2020 [34]	Invasive	PIRA	No	No	64	33 ± 8	100	End of CT regimen	3D, Toshiba Artida SSH-880CV	N.A.	N.A.	N.A.	N.A.
Xu, 2021 [35]	Newly diagnosed	EPI	No	No	95	53 ± 9	100	≅12 months after CT initiation	3D, GE Vivid E9 EchoPAC Software	62 ± 5	61 ± 5	20.8 ± 2.3	17.7 ± 1.8
Attar, 2022 [36]	67% with breast cancer	Probably EPI	No	No	60	43 ± 12	68	End of CT regimen (≅ 6 months)	2D, GE Vivid E9 GE AFI Software	59 ± 6	51 ± 6	21.1 ± 1.8	18.6 ± 2.6
Laufer-Perl, 2022 [37]	N.A.	DOX	25%	45%	40	50 ± 13	100	≅3 months post-CT stop	2D, GE Vivid S70 TomTec Software	60 ± 1	59 ± 2	21.5 ± 2	19.7 ± 1.8
El-Sherbeny, 2023 [38]	HER2 negative	Probably EPI	No	No	66	47 ± 8	100	≅9 months post-CT stop	2D and 3D, GE Vivid E9, GE AFI and 4D RVQ Software	62 ± 5	57 ± 5	21.7 ± 1.3	19.0 ± 1.3
Ghaznawie, 2023 [39]	85% with breast cancer	DOX or EPI	No	No	34	48 ± 11	94	End of CT regimen	2D, GE Vivid E95 EchoPAC Software	N.A.	N.A.	N.A.	N.A.
Giang, 2023 [40]	Newly diagnosed, HER2 negative	DOX	No	No	351	58 ± 8	100	≅3 weeks post-CT stop	2D, Philips Affiniti QLAB15.0 Software	64 ± 5	62 ± 6	18.7 ± 1.4	17.8 ± 1.3
Fawzy, 2024 [41]	Non-metastatic	DOX or EPI	50%	No	101	49 ± 10	100	≅3 months after CT initiation	2D, GE Vivid 5 or 7 EchoPAC Software	60 ± 3	57 ± 3	18.5 ± 1.5	16.7 ± 1.9
Gorgiladze, 2024 [42]	Newly diagnosed	DOX or EPI	No	No	28	49 ± 12	100	End of 4-cycle CT regimen (≅ 4 months)	3D, GE Vivid E9 EchoPAC Software	63 ± 4	65 ± 3	21.2 ± 2.1	18.6 ± 2.6
Rossetto, 2024 [43]	Non-metastatic	DOX	18%	19%	83	55 ± 11	100	≅12 months after CT initiation	2D, GE Vivid E95 EchoPAC Software	62 ± 4	58 ± 5	20.5 ± 1.7	18.4 ± 2.0

**Table 2 cancers-16-03883-t002:** Right Ventricular Function Before and After Anthracycline Chemotherapy Across the Studies Included in the Systematic Review and Meta-Analysis. RV GLS: right ventricle global longitudinal strain, RV FWLS: right ventricle free wall longitudinal strain, TAPSE: Tricuspid Annular Plane Systolic Excursion, FAC: Fractional Area Change, S’: Systolic Peak Velocity, PAPs: Pulmonary Artery Systolic Pressure, RVEF: Right Ventricular Ejection Fraction, N.A.: not available.

Author, Publication Year	Sample Size	RV GLS (%)		RV FWLS (%)		TAPSE		FAC (%)		S’		PAPs		RVEF (%)	
		Before	After	Before	After	Before	After	Before	After	Before	After	Before	After	Before	After
Boczar, 2016 [29]	49	N.A.	N.A.	16.2 ± 6.3	13.8 ± 6.8	N.A.	N.A.	48 ± 9	42 ± 9	N.A.	N.A.	N.A.	N.A.	N.A.	N.A.
Chang, 2016 [30]	35	N.A.	N.A.	22.5 ± 5.0	16.9 ± 7.3	19.4 ± 4.7	12.2 ± 6.2	61 ± 12	56 ± 7	14.8 ± 3.5	12.8 ± 6.7	15 ± 5	18 ± 8	N.A.	N.A.
Arciniegas Calle, 2018 [31]	66	22.7 ± 5.5	19.0 ± 5.9	N.A.	N.A.	N.A.	N.A.	N.A.	N.A.	N.A.	N.A.	N.A.	N.A.	N.A.	N.A.
Wang, 2018 [32]	36	28.6 ± 1.8	15.2 ± 2.7	N.A.	N.A.	23.0 ± 2.2	21.9 ± 3.8	50 ± 4	48 ± 4	18.9 ± 1.5	18.2 ± 1.3	N.A.	N.A.	N.A.	N.A.
Anqi, 2019 [33]	40	20.3 ± 6.0	18.8 ± 7.1	N.A.	N.A.	23.1 ± 4.4	20.4 ± 4.0	52 ± 9	48 ± 9	14.3 ± 2.8	14.1 ± 2.5	N.A.	N.A.	N.A.	N.A.
Wang, 2020 [34]	64	28.4 ± 2.4	25.4 ± 2.4	N.A.	N.A.	24.6 ± 2.1	24.3 ± 1.9	53 ± 5	53 ± 5	25.7 ± 2.0	25.7 ± 1.9	N.A.	N.A.	N.A.	N.A.
Xu, 2021 [35]	95	21.5 ± 3.2	18.6 ± 2.6	25.8 ± 2.9	21.6 ± 2.5	21.6 ± 2.4	19.9 ± 2.0	44 ± 4	43 ± 3	13.1 ± 2.3	12.4 ± 2.0	25 ± 3	27 ± 3	56 ± 3	53 ± 3
Attar, 2022 [36]	60	22.9 ± 2.0	18.5 ± 2.8	25.8 ± 3.0	20.3 ± 3.8	24.7 ± 2.3	20.6 ± 2.1	54 ± 4	46 ± 6	12.0 ± 1.5	11.9 ± 1.4	N.A.	N.A.	N.A.	N.A.
Laufer-Perl, 2022 [37]	40	26.8 ± 4.7	21.5 ± 4.4	28.9 ± 5.1	25.6 ± 5.9	25.0 ± 3.0	N.A.	N.A.	N.A.	N.A.	N.A.	26 ± 6	N.A.	N.A.	N.A.
El-Sherbeny, 2023 [38]	66	23.2 ± 0.8	21.6 ± 1.1	25.1 ± 1.5	23.2 ± 1.7	18.7 ± 1.8	18.0 ± 1.0	45 ± 5	43 ± 5	14.1 ± 1.5	13.2 ± 1.9	14 ± 4	16 ± 4	52 ± 4	50 ± 5
Ghaznawie, 2023 [39]	34	N.A.	N.A.	23.3 ± 3.5	18.4 ± 6.0	N.A.	N.A.	N.A.	N.A.	N.A.	N.A.	N.A.	N.A.	N.A.	N.A.
Giang, 2023 [40]	351	23.6 ± 2.8	23.4 ± 2.9	27.6 ± 3.7	27.5 ± 4.1	N.A.	N.A.	N.A.	N.A.	N.A.	N.A.	N.A.	N.A.	N.A.	N.A.
Fawzy, 2024 [41]	101	23.1 ± 1.0	20.4 ± 2.8	25.3 ± 1.1	22.5 ± 3.0	N.A.	N.A.	N.A.	N.A.	12.8 ± 0.9	11.9 ± 1.2	N.A.	N.A.	N.A.	N.A.
Gorgiladze, 2024 [42]	28	N.A.	N.A.	25.2 ± 2.9	21.4 ± 4.4	24.4 ± 3.0	25.4 ± 2.5	N.A.	N.A.	11.0 ± 7.0	14.2 ± 2.0	N.A.	N.A.	N.A.	N.A.
Rossetto, 2024 [43]	83	22.2 ± 2.5	21 ± 4.1	26.5 ± 3.8	24.6 ± 3.6	22.4 ± 2.8	22.3 ± 2.7	50 ± 8	46 ± 5	13.5 ± 2.0	11.6 ± 7.4	N.A.	N.A.	59 ± 5	53 ± 6

## Data Availability

The data underlying this article will be shared on reasonable request to the corresponding author.

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
