# Peer review of "Anthracycline-Induced Subclinical Right Ventricular Dysfunction in Breast Cancer Patients: A Systematic Review and Meta-Analysis"

_cancers, 2024, doi:10.3390/cancers16223883_

Round 1
Reviewer 1 Report
Comments and Suggestions for Authors
Major Comments: The present manuscript is a meta-analysis to evaluate the effects of anthracyclines on echocardiographic parameters of the right ventricle, explicitly focusing on RV global longitudinal strain (RV GLS) and RV free-wall longitudinal strain (RV FWLS). The study is well-designed and adheres to the Preferred Reporting Items for Systematic Reviews and Meta-Analyses (PRISMA) guidelines, ensuring methodological rigor. Furthermore, this systematic review has been registered with PROSPERO under the identifier CRD42024591588. The quality assessment of the studies included in this analysis yielded scores ranging from 6 to 9 on the Newcastle-Ottawa Scale (NOS), categorizing them as fair to good quality, but no studies were excluded due to suboptimal quality.
Despite the significant scientific interest in this topic, as well as the manuscript's clarity and coherence, the findings do not provide definitive evidence regarding whether right ventricular dysfunction arises as a secondary consequence of left ventricular dysfunction or anthracyclines treatment. To address this analytical limitation, the authors conducted meta-regressions indicating no significant correlation between post-chemotherapy left ventricular ejection fraction (LVEF) and left ventricular global longitudinal strain (LV GLS) with subclinical right ventricular dysfunction. Consequently, the authors suggest that anthracyclines may exert direct cardiotoxic effects on the right heart.
However, an essential limitation of this study is the need for more detail regarding the treatment regimens involving anthracyclines, such as treatment duration, the number of cycles administered, the cumulative dosage, and any necessary modifications to dosing. Left ventricular cardiotoxicity is dose-dependent, which raises important questions about whether right ventricular cardiotoxicity is similarly influenced by the dosage of anthracyclines.
Minor Comments: While an abbreviation list is included, the authors should ensure the standardized use of full descriptions for all abbreviations as they first appear in the text. It is noted that several abbreviations lack their complete forms in the body of the manuscript, which could potentially confuse readers.
Author Response
Major Comments: The present manuscript is a meta-analysis to evaluate the effects of anthracyclines on echocardiographic parameters of the right ventricle, explicitly focusing on RV global longitudinal strain (RV GLS) and RV free-wall longitudinal strain (RV FWLS). The study is well-designed and adheres to the Preferred Reporting Items for Systematic Reviews and Meta-Analyses (PRISMA) guidelines, ensuring methodological rigor. Furthermore, this systematic review has been registered with PROSPERO under the identifier CRD42024591588. The quality assessment of the studies included in this analysis yielded scores ranging from 6 to 9 on the Newcastle-Ottawa Scale (NOS), categorizing them as fair to good quality, but no studies were excluded due to suboptimal quality.
Despite the significant scientific interest in this topic, as well as the manuscript's clarity and coherence, the findings do not provide definitive evidence regarding whether right ventricular dysfunction arises as a secondary consequence of left ventricular dysfunction or anthracyclines treatment. To address this analytical limitation, the authors conducted meta-regressions indicating no significant correlation between post-chemotherapy left ventricular ejection fraction (LVEF) and left ventricular global longitudinal strain (LV GLS) with subclinical right ventricular dysfunction. Consequently, the authors suggest that anthracyclines may exert direct cardiotoxic effects on the right heart.
However, an essential limitation of this study is the need for more detail regarding the treatment regimens involving anthracyclines, such as treatment duration, the number of cycles administered, the cumulative dosage, and any necessary modifications to dosing. Left ventricular cardiotoxicity is dose-dependent, which raises important questions about whether right ventricular cardiotoxicity is similarly influenced by the dosage of anthracyclines.
A: Thank you very much Reviewer. We are delighted that you appreciated our work. Your suggestion was very helpful, as it allowed us to test the dose-dependency hypothesis in RV cardiotoxicity:
- We have incorporated the text with Supplementary Table 1, which outlines the specifics of the chemotherapy regimen used in the various studies. Overall, the pooled cumulative DOX-equivalent dose was found to be 234.6± 2 mg/m².
- We conducted a meta-regression analysis examining the relationship between the cumulative DOX-equivalent dose and the effect sizes of RV GLS and RV FWLS (expressed as SMD). This analysis aimed to evaluate the influence of anthracycline dosage on the incidence of subclinical RV dysfunction. The analysis did not reveal a significant relationship between the SMD of RV GLS and RV FWLS and the cumulative DOX-equivalent dose (coefficient: 0.0002 ,p =0.989 for RV GLS; coefficient: 0.0242, p = 0.876 for RV FWLS, respectively). A specific comment was added in the discussion section “Interestingly, our meta-regression suggests that RV cardiotoxicity does not appear to be dose-dependent on anthracycline exposure, in contrast to LV cardiotoxicity. This supports the hypothesis of a distinct underlying damage mechanism. However, it is important to note that all studies included in our analysis reported cumulative DOX-equivalent doses ranging from 193.1 to 258.4 mg/m²—substantially below the critical thresholds (>400-450 mg/m²)—which may limit our capacity to detect significant dose-dependent effects at higher doses”
Minor Comments: While an abbreviation list is included, the authors should ensure the standardized use of full descriptions for all abbreviations as they first appear in the text. It is noted that several abbreviations lack their complete forms in the body of the manuscript, which could potentially confuse readers.
A: Thank you very much Reviewer. The manuscript has been revised accordingly with your indication.
Reviewer 2 Report
Comments and Suggestions for Authors In this manuscript, the authors performed a meta-analysis of the effects of anthracycline chemotherapy in breast cancer on markers of right ventricular function, including right ventricular global longitudinal and free wall strain, right ventricular ejection fraction, and other measures of basal right ventricular function. The findings demonstrate that these markers of right ventricular function decline with treatment, and through a meta-regression, they are not correlated with changes in left ventricular ejection fraction, suggesting that the effects of anthracycline chemotherapy on right ventricular function may be not only dependent on the onset of left ventricular dysfunction. Overall, the authors have done an excellent job in conducting this study. The question of what are the effects of anthracycline chemotherapy on right ventricular function as well as the prognostic implications of these changes are not well known, and this meta-analysis makes the case clearly that right ventricular function is impaired by anthracycline exposure. The meta-analysis is generally well conducted, with a solid search and inclusion strategy and good methodological approach. I propose the following suggestions to further improve the quality of the study: 1. Either an analysis of the effect of HER2 treatment and radiotherapy on RV GLS should be included or it should discussed why this was not possible in the discussion section. The mechanisms of cardiac toxicity of these treatments are distinct from anthracycline toxicity and so may potentially have differing effects on right ventricular function. Understandably, only a minority of patients in the studies analyzed received these treatments, so definitive conclusions may not be able to be made, but regardless this point should at least be addressed. 2. Reference 17 - the abstract is English but article text is in Chinese. I am unable to find a version of the article published in English. Either the study inclusion criteria should be updated to specify that articles with English language abstracts will be included, or this study should be removed from analysis. 3. Add the Newcastle-Ottawa Scale assessment for each study either to table 1 or include in a supplemental figure.Author Response
In this manuscript, the authors performed a meta-analysis of the effects of anthracycline chemotherapy in breast cancer on markers of right ventricular function, including right ventricular global longitudinal and free wall strain, right ventricular ejection fraction, and other measures of basal right ventricular function. The findings demonstrate that these markers of right ventricular function decline with treatment, and through a meta-regression, they are not correlated with changes in left ventricular ejection fraction, suggesting that the effects of anthracycline chemotherapy on right ventricular function may be not only dependent on the onset of left ventricular dysfunction. Overall, the authors have done an excellent job in conducting this study. The question of what are the effects of anthracycline chemotherapy on right ventricular function as well as the prognostic implications of these changes are not well known, and this meta-analysis makes the case clearly that right ventricular function is impaired by anthracycline exposure. The meta-analysis is generally well conducted, with a solid search and inclusion strategy and good methodological approach.
A: Thank you very much Reviewer. We are delighted that you appreciated our work
I propose the following suggestions to further improve the quality of the study:
- Either an analysis of the effect of HER2 treatment and radiotherapy on RV GLS should be included or it should discussed why this was not possible in the discussion section. The mechanisms of cardiac toxicity of these treatments are distinct from anthracycline toxicity and so may potentially have differing effects on right ventricular function. Understandably, only a minority of patients in the studies analyzed received these treatments, so definitive conclusions may not be able to be made, but regardless this point should at least be addressed.
A: Thank you very much, Reviewer, for highlighting this important point, which has now been addressed in the limitations section as follows:
“Unfortunately, only few studies included subjects who underwent radiotherapy and/or concurrent anti-HER2 therapy, and especially did not provide separate data for patients with and without these concomitant therapies. This limitation hindered our ability to assess any potential differential effects of these treatments on the development of subclinical RV dysfunction.”.
- Reference 17 - the abstract is English but article text is in Chinese. I am unable to find a version of the article published in English. Either the study inclusion criteria should be updated to specify that articles with English language abstracts will be included, or this study should be removed from analysis.
A:Thank you, Reviewer, for pointing out this oversight. We have removed the requirement in the inclusion criteria that the text must necessarily be in English. Indeed, despite the fact that Manuscript Ref 17 was not in English, its comprehension was not limiting, and therefore, it deserves inclusion.
- Add the Newcastle-Ottawa Scale assessment for each study either to table 1 or include in a supplemental figure.
A: Thank you very much, Reviewer. The manuscript has been revised in accordance with your suggestion, and we have added a supplemental Table 2 with the NOS scale.
Round 2
Reviewer 2 Report
Comments and Suggestions for Authors
My thanks to the authors in addressing my recommendations to strengthen the content of the manuscript. I believe the changes have adequately addressed my concerns.
I am not convinced that of the addition of the dose-dependency analysis is a positive addition to the manuscript. The analysis required inference of the type of anthracycline given based on cumulative doses reported which is a potential significant source of error. Additionally, as noted in the discussion, the cumulative anthracycline doses are fairly low which limits the sensitivity of this analysis. From this analysis, it is difficult to conclude if it is that RV toxicity is not dose dependent, or rather that the available data is not sufficient to prove an association. As a positive control, the analysis could be repeated with markers of LV function (GLS / EF) to see if dose-dependence can be demonstrated with the data.